# Territorial Strategy of Medical Units for Addressing the First Wave of the COVID-19 Pandemic in the Metropolitan Area of Mexico City: Analysis of Mobility, Accessibility and Marginalization

**DOI:** 10.3390/ijerph19020665

**Published:** 2022-01-07

**Authors:** Mateo Carlos Galindo-Pérez, Manuel Suárez, Ana Rosa Rosales-Tapia, José Sifuentes-Osornio, Ofelia Angulo-Guerrero, Héctor Benítez-Pérez, Guillermo de Anda-Jauregui, Juan Luis Díaz-de-León-Santiago, Enrique Hernández-Lemus, Luis Alonso Herrera, Oliva López-Arellano, Arturo Revuelta-Herrera, Rosaura Ruiz-Gutiérrez, Claudia Sheinbaum-Pardo, David Kershenobich-Stalnikowitz

**Affiliations:** 1Centro Regional de Investigaciones Multidisciplinarias, Universidad Nacional Autónoma de México, Cuernavaca 62209, Mexico; 2Instituto de Geografía, Universidad Nacional Autónoma de México, Ciudad de Mexico 04510, Mexico; anarosa@igg.unam.mx; 3Instituto Nacional de Ciencias Médicas y Nutrición Salvador Zubirán, Ciudad de Mexico 14080, Mexico; jose.sifuenteso@incmnsz.mx (J.S.-O.); kesdhipa@yahoo.com (D.K.-S.); 4Secretaría de Educación, Ciencia, Tecnología e Innovación, Gogobierno de la Ciudad de México, Ciudad de Mexico 06010, Mexico; ofelia.angulo@gmail.com (O.A.-G.); juan.diaz@cdmx.gob.mx (J.L.D.-d.-L.-S.); rosaura@ciencias.unam.mx (R.R.-G.); 5Dirección General de Cómputo y de Tecnologías de Información y Comunicación, Universidad Nacional Autónoma de México, Ciudad de Mexico 04510, Mexico; hector.benitez@iimas.unam.mx; 6Departamento de Genómica Computacional, Instituto Nacional de Medicina Genómica, Ciudad de Mexico 14610, Mexico; gdeanda@inmegen.edu.mx (G.d.A.-J.); ehernandez@inmegen.gob.mx (E.H.-L.); 7Instituto de Investigaciones Biomédicas, Universidad Nacional Autónoma de México, Ciudad de Mexico 04510, Mexico; metil@hotmail.com; 8Secretaría de Salud de la Ciudad de México, Ciudad de Mexico 06900, Mexico; olivalopez@gmail.com (O.L.-A.); arrevuelta@gmail.com (A.R.-H.); 9Gobierno de la Ciudad de México, Ciudad de Mexico 06000, Mexico; claudia.sheinbaum@cdmx.gob.mx

**Keywords:** COVID hospitals, contagion, mobility, accessibility, Mexico City Metropolitan Area

## Abstract

Background. The COVID-19 pandemic has caused an exponential increase in the demand for medical care worldwide. In Mexico, the COVID Medical Units (CMUs) conversion strategy was implemented. Objective. To evaluate the CMU coverage strategy in the Mexico City Metropolitan Area (MCMA) by territory. Materials. The CMU directory was used, as were COVID-19 infection and mobility statistics and Mexican 2020 census information at the urban geographic area scale. The degree of urban marginalization by geographic area was also considered. Method. Using descriptive statistics and the calculation of a CMU accessibility index, population aggregates were counted based on coverage radii. In addition, two regression models are proposed to explain (1) the territorial and temporal trend of COVID-19 infections in the MCMA and (2) the mobility of the COVID-infected population visiting medical units. Results. The findings of the evaluation of the CMU strategy were (1) in the MCMA, COVID-19 followed a pattern of contagion from the urban center to the periphery; (2) given the growth in the number of cases and the overload of medical units, the population traveled greater distances to seek medical care; (3) after the CMU strategy was evaluated at the territory level, it was found that 9 out of 10 inhabitants had a CMU located approximately 7 km away; and (4) at the metropolitan level, the lowest level of accessibility to the CMU was recorded for the population with the highest levels of marginalization, i.e., those residing in the urban periphery.

## 1. Introduction

The first outbreak of 27 cases of COVID-19 pneumonia was recorded in Wuhan, China, on 31 December 2019. Since this outbreak, the disease rapidly disseminated worldwide [1,2], and the number of infections and deaths increased exponentially throughout the world in the first quarter of 2020. This was recorded by countries such as the USA [3], Spain [4], Italy [5] and England and Wales [6]. The World Health Organization (WHO) declared a public health emergency of international importance on January 30, 2020, and the COVID-19 pandemic was declared on 11 March 2020.

In Mexico, the first case of COVID-19 was identified on 27 February 2020, and the first COVID-19 related death was recorded on March 18. Five days later, on 23 March 2020, Mexico announced the start of the quarantine, and the decision was made to close schools and suspend nonessential activities. A week later, on 2 April 2020, a state of emergency was declared throughout the country. During this initial phase of the pandemic, Mexico ranked third in the world in the number of recorded infections and deaths, behind the USA and Brazil.

By 30 April 2020, the number of confirmed infections in Mexico totaled 19,224 cases and 1859 deaths [7]. In the MCMA, the figures were 8422 infections and 570 deaths, meaning that 44% of infections and 31% of total deaths were concentrated in the largest and most populated urban area of the country, as was to be expected given the research into the relationship between a city’s size [8], population density and mobility [9,10,11], and the increase in the probability of contagion.

Given the level of impact of the disease (number of infections and deaths) and its constant growth, the Mexican government designed and implemented a national hospital-coverage strategy, through which dedicated COVID hospitals were designated.

The general objective of this strategy was to convert hospitals in order to allow them to cater exclusively to COVID-19 cases, regardless of the patient’s eligibility (anyone could access medical care). The conversion consisted of adapting hospital care, taking into account the availability of intensive-care beds for critical patients and the number of hospital beds for noncritical patients. In summary, the aim was to modify the regular functionality of some medical units, to change the usual provision of services and direct them toward caring for patients with suspected or diagnosed COVID-19. The strategy included new, authorized or provisional hospitals, and (in some cases) the restructuring could be expanded up to 100% of capacity depending on the demand [12].

In the first stage of hospital restructuring (baseline installed capacity), some of the criteria that were used to select units considered only hospitals that (1) were located in urban areas, according to population density; (2) had four or more beds in the intensive care units for adults; (3) had the human resources to provide direct COVID-19 care: physicians and nurses (general and specialists), radiology and imaging technicians, laboratory technicians and respiratory therapists; (4) had considerable areas available for the care of COVID-19 patients: triage, emergency services, an adult intensive-care unit, hospitalization, a pharmacy, a laboratory and imaging; and (5) had the following medical equipment available and in operation: ventilators, monitors, infusion pumps, portable X-ray equipment, crash carts and ultrasound.

This strategy was adopted in order to mobilize the greatest possible quantity of human and material resources to cater to the growing number of cases at the local, regional and national levels. From a territorial perspective, the goal was to increase access to health services for medical care by the COVID-19-infected population [13,14]. At the federal level, the health institutions involved in this strategy were the Mexican Social Security Institute (referred to as the IMSS), the Department of Health (Ssa), the Social Security and Services Institute for State Workers (ISSSTE), the Ministry of National Defense (Sedena) and the Secretariat of the Navy (Semar) [14]. At the local level, the following organizations were incorporated into the COVID Medical Units (CMU): the Secretariat of Health of Mexico City (referred to as the Sedesa), the Health Institute of the State of Mexico (ISEM) and the Social Security Institute of the State of Mexico and its Municipalities (ISSEMyM). In addition, three temporary centers were established in CDMX [15,16,17].

Altogether, by 30 April 2020, 83 medical units in the MCMA were restructured, making 15,460 hospital beds available for the treatment of COVID-19 cases. The total number of beds is taken to be the potential number of patients (Table 1). The distribution of COVID hospitals follows a center-periphery territorial pattern, starting from what we call the metropolitan health center [18], as can be seen in Figure 1.

Additionally, given the continuous increase in the number of infections and deaths, the federal government decided to sign its first cooperation agreement with private institutions (30 April 2020), an initiative which was called “Together for Health” [19]; months later, a second agreement was signed (17 November 2020) to convert private hospitals to treat COVID-19 exclusively [20].

Previous research has found that the degree of vulnerability to COVID-19 is related to sociodemographic and economic characteristics [21] that include location factors, including city characteristics [22]. Conditions associated with poverty have been found to increase the probability of contagion [23] and decrease the ability to confront the pandemic [24]. Furthermore, public policy designed for the average population is less effective for minority and lower-income groups faced by accentuated pre-existing imbalances such as social inequities, racial/ethnic gaps, and aging [25].

Galindo and Suárez [26] reported a hypothetical scenario of the extraordinary demand for health services that a pandemic could cause in Mexico City. They concluded that 70% of the population of the MCMA would not receive medical care, and that the city’s most marginalized sectors would be the worst affected. This research recommended that, in addition to reconverting medical units to increase response capacity and treat infections, a strategy related to the location and coverage of hospitals in the city should be designed and implemented.

This document examines the health policy implemented by the federal and local governments in Mexico City (CDMX) and its metropolitan area as regards the restructuring of hospitals to face the health emergency. To do so, we analyze the spatial pattern of contagion, the impact of the overload of medical units on the distances traveled by the COVID-19-infected population to seek medical attention, as well as accessibility to medical units in relation to the marginalization level of the population. Marginalization is measured using principal component analysis, using socioeconomic variables that include income, education, housing characteristics and social security.

In the following sections we present the study area, data sources as well as the analytical and statistical methods used. Subsequently, we present the results of our analyses, followed by a discussion section with our main conclusions.

## 2. Materials and Methods

### 2.1. Study Area

The MCMA comprises 16 municipalities in CDMX, 59 municipalities in the State of Mexico and one municipality in the State of Hidalgo [27]. As of 2020, the MCMA had 21.8 million inhabitants, which is equivalent to 17% of the total population of Mexico.

### 2.2. Mexico’s Health Care System

Mexico’s Health Care System has three components: (1) social security, (2) social assistance, and (3) the private health service. The first and second are organized at two administrative levels: (1) federal and (2) local. As regards social security, the federal level is in charge of providing medical care to workers from the formal private sector (IMSS) and federal workers (ISSSTE, Mexican Petroleum (Pemex), the Army, and the Navy). At the local level, social security is the responsibility of state governments, which provide medical care to their workers (State Health Institutes). As regards social assistance, federal-level bodies are responsible for providing medical care to freelance workers, the underemployed, the unemployed, non-workers and their familiars (SSA). The healthcare system distinguishes and organizes hospitals on three levels of medical attention: primary, secondary, and tertiary.

### 2.3. Demographic and Population Mobility Data

Data as of 30 April 2020 on (1) the number of infections and deaths from COVID-19 and (2) visits to medical units due to COVID-19 were used. The source was the Mexican Department of Health. Additionally, the COVID hospital directory for the MCMA was used. For the evaluation of the hospital strategy in the MCMA, the base input was the number of hospital beds available per hospital, which is assumed to be the full hospital capacity.

Additionally, the total population was based on the 2020 population census of Mexico, divided into urban geographical areas (TRACTs or Census Tracts) [28]. It is emphasized that the total population of the MCMA was taken into account because the health measures implemented worldwide considered that, given the lack of knowledge of this new disease, its virulence and the presence of premorbid conditions, the entire population was vulnerable to COVID-19 contagion and illness.

### 2.4. Analytical and Statistical Methods

The territorial evaluation of the COVID hospital strategy in the MCMA was done in four stages:

#### 2.4.1. Identification of Infection Trends by Territory

Descriptive statistical methods were used for the numbers of infections and deaths from COVID-19. A first regression model was constructed to explain the increase and territorial distribution of the number of positive COVID-19 cases (*Y*) by borough/municipality as of 30 April 2020. The independent variables (*X*_1_ + *X*_2_ + *X*_3_) were, respectively: (1) the logarithm of the number of days elapsed between metropolitan case 0 and the first case in borough/municipality; (2) the distance (in kilometers by road) from each municipal/borough seat to the metropolitan center; and (3) urban population density.

#### 2.4.2. Recognition of Patterns of Mobility toward Medical Units

Next, the mobility of the population with COVID-19 infection to medical units was examined. Statistics were reviewed, and mobility flows were mapped. In addition, the average distance traveled to medical units per health institution was calculated.

#### 2.4.3. Evaluation of the Hospital Strategy by Territory

For the territorial evaluation of the COVID hospital strategy itself, coverage areas with a radius of 10 km were drawn for each hospital. These distance ranges were chosen based on the following assumptions: (1) based on a rational decision, any person (as a first option) will go to the nearest hospital for care [29]; (2) the average distances (within a city)—there was no established consensus regarding the distance thresholds (minimum or maximum) to be covered. Thresholds vary among regions, countries (developed, developing) and cities (surface area, population size) [29]—that a person was willing to travel to get to a hospital (under normal conditions) by public or private transport were considered to be within a range of 10 km [30,31]; and (3) practically the entire urban area of the MCMA was covered by these coverage areas.

Additionally, new coverage areas were drawn from the center of the MCMA toward its periphery, to a radius of 10 km. Next, based on data from the 2020 population census of Mexico [28], the total population for the established coverage areas was counted and aggregated at the urban-tract level (census tracts). Using this procedure, the potential demand was estimated and compared, as was the coverage of COVID hospitals at the local and metropolitan levels.

#### 2.4.4. Measurement of Accessibility to COVID Hospitals

An index of the accessibility of COVID hospitals was calculated at the urban-tract level [32,33,34] to estimate by territory the ease with which the population could reach a COVID hospital as a function not only of distance but of the coverage capacity (number of hospital beds).

In general terms, accessibility refers to the ease with which a person can reach urban opportunities (destinations), such as workplaces, schools, or hospitals [35]. This definition implicitly gives greater weight to the separation between origins and destinations, measured in time or distance. Contributions to this concept recognize that accessibility is the potential opportunity for reaching/arriving at a destination depending on transportation options, in which the geographical distribution of the population (demand) and destinations (supply) must be considered [36,37].

Based on this definition, the accessibility of COVID hospitals is understood as the ease with which people (demand) can reach a COVID hospital for treatment; however, the capacity of each hospital (supply of beds) to care for a given number of cases must also be considered [38,39,40,41]. Thus, the equation used to calculate the accessibility index was the following:(1)$$AI_{i}=\underset{j}{\sum }\left(N_{j}/P_{i}\right)\cdot d_{ij}^{-1}$$
where *AI_i_* = the COVID hospital accessibility index for TRACT*i*; *N_j_* = the number of hospital beds in COVID hospital *j* in year *x*; *P_i_* = the total population of TRACT*i* in census year *x*; *d_ij_* = distance in kilometers by road from the centroid of TRACT*i* to COVID hospital *j*. The distance was calculated from the centroid of each urban tract to each of the COVID hospitals using Geographic Information Systems. The index thus obtained was classified into five quintiles and mapped.

Subsequently, the accessibility index was correlated with urban marginalization, as reported by tract [42]. To determine the degree of spatial inequality in health services among socioeconomic groups, urban marginalization was chosen as it is an index of basic welfare deficiencies. Among the indicators considered in this index are housing conditions, average educational level, and lack of accessibility to health services.

A second regression model was constructed to statistically support and detect the effect of the spatial imbalance between the supply of COVID hospitals and users. Here, the dependent variable (*Y*) was the logarithm of the number of visits to medical units due to possible COVID-19 infection (on 30 April 2020). The independent variables (*X_1_* + *X_2_* +... + *X_n_*) were, respectively: (1) the urban marginalization index; (2) the percentage of the population with some disability; (3) the percentage of the population that was economically dependent on others; (4) the distance (in kilometers by road) from each municipal and borough seat to the metropolitan center; and (5) the COVID hospitals accessibility index (weighted by the total population of the borough/municipality).

## 3. Results

The results determined using the methods described above are presented in the following four subsections:

### 3.1. COVID-19 Infection Trend by Territory

The speed of transmission and number of COVID-19 infections in the MCMA are shown in Figure 2. What can be observed is a territorial pattern moving outward from the urban center to the periphery; practically all boroughs and most of the surrounding municipalities registered cases during the first month of the pandemic. The furthest peripheral areas registered cases in the second and third months.

The regression model indicates that the number of infections at the cutoff date was higher in those municipalities in which the first case occurred closer to the date of the metropolitan index case. All other things being equal, the model shows that the municipalities closest to the urban center and those with a higher population density had a larger number of cases. These variables are related to the dynamics of the city in terms of workflows and trip attractors, which increase as the distance from the center decreases. In this way, the coefficients may indicate that the number of positive cases of COVID-19 will be higher in places with greater social interaction (Table 2). These also match central locations where a large hospital infrastructure is found, implying a high accessibility to COVID hospitals in the initial stages of the pandemic, which will necessarily decrease as the number of positive cases spreads to the periphery of the city.

### 3.2. Mobility of the Population to Medical Units

The center-periphery territory pattern of the speed of transmission and number of COVID-19 infections in the MCMA was reflected in the mobility trends of the population seeking medical attention for possible contagion. In the MCMA, more than 24,500 visits to medical units due to COVID-19 were recorded. The health institution that received the highest number of patients or visits was the Ssa; in second place was the IMSS; in third place were medical units in the private sector; and in fourth place was the ISSSTE. As of 30 April 2020, these four institutions received 97.5% of the visits to medical units due to COVID-19 (Table 3).

In terms of territory, the medical units of the different health institutions attracted people from every state in the country, but the bulk of the visits were made by MCMA residents (Figure 3, Figure 4, Figure 5 and Figure 6).

The trend of a center-periphery pattern of mobility for seeking medical care can be observed. The origins are the peripheral boroughs and suburbs, and the preferred destination is the city center, mainly along the corridor that concentrates the bulk of the hospital offerings in the MCMA. This corridor extends from the Gustavo A. Madero municipality (La Raza National Medical Center, of the IMSS) in the north of CDMX through the Cuauhtémoc and Benito Juárez municipalities until it reaches the south, the Tlalpan municipality (hospital zone). On the other hand, the average Euclidean distance to reach a medical unit of one of the four main health institutions is shown in Figure 7 and Table 4. The figure shows that central municipalities of the MCMA register the lowest average distance, which is explained by the greater agglomeration of medical units in this area of the MCMA. On the other hand, toward the metropolitan periphery, the average distance increases gradually.

### 3.3. Territorial Evaluation of the COVID Hospital Strategy

Based on the coverage areas established for each hospital (up to a radius of 10 km), at the local level, 9 out of 10 inhabitants of the MCMA have a COVID hospital within the threshold of 7 km (Figure 8 and Figure 9).

To evaluate coverage at the metropolitan level based on new areas of coverage (within a 10 km range), the concentration of hospital beds and the distribution of the total urban population were compared, but this time as a function of the distance to the metropolitan center. Figure 10 shows that hospital beds are clearly more heavily distributed in the center of the MCMA, and that distribution decreases progressively toward the periphery. In comparison, the population distribution also follows a center-periphery pattern, but an inverted one; it is lower in the center, increases in the second perimeter area and decreases toward the periphery.

For the whole of the MCMA, 43% of the hospitals and 57% of the beds are concentrated within a radius of 10 km from the metropolitan health subcenter. When a 20 km radius is considered, the percentage rises to 77% of hospitals and 85% of beds (Table 5). Regarding the urban population, 20% is concentrated within a radius of 10 km from the center; when a radius of 20 km is considered, the percentage increases to 61%.

This result reflects a spatial mismatch between the locations of COVID hospitals (supply of beds) and the population area, which led to the movement of infected patients from the periphery toward the center of the MCMA to seek medical attention (as shown in the previous section).

If the content of Table 4 is compared with the results of this section, it can be observed that although 9 out of 10 inhabitants of the MCMA can reach a COVID hospital within 7 km, the average distance between patients and hospitals is twice as great.

### 3.4. Accessibility to COVID Hospitals

According to the range of accessibility to COVID hospitals, 36% of the population of the MCMA has a very low degree of accessibility, and another 25% has a low degree, representing almost 12.76 million inhabitants in total. At the other extreme, 20% of the population of the MCMA is included in the ranges of high and very high accessibility (Figure 11 and Figure 12).

The association of the accessibility index with the degree of urban marginalization shows that marginalization increases with lower accessibility (Table 6 and Figure 13).

Using an additional variable, the range of accessibility, the average distances between the closest COVID hospital and the urban health subcenter were compared. The greater the accessibility, the shorter the distance to COVID hospitals. When the average distance was compared with the degree of marginalization, it was seen that the greater the urban marginalization, the more the distance from COVID hospitals increases (Table 7). This implies that populations affected by COVID with fewer resources have lower accessibility to health services, requiring them to travel greater distances to receive hospital care during the pandemic.

The regression model (Table 8) was statistically significant and validated the hypothesis of a spatial mismatch between population area and the location/coverage (supply of hospital beds) of COVID hospitals, requiring demand, i.e., those needing medical care (COVID-19 cases) to move to the center of the MCMA in their search for it.

In the regression model, the coefficients may indicate that a greater number of trips to seek medical care (at the cutoff date) is associated with a lower index of marginalization and a shorter road distance to the metropolitan center. This can be explained by the spread of COVID-19 in the surrounding boroughs and municipalities (the near periphery).

However, it should be noted that the borough/municipality scale does not allow the identification of inland areas of high marginalization, being the areas that reported the greatest number of COVID-19 infections. Reality showed that the southern periphery of CDMX, which had the highest level of marginalization in the MCMA, was one of the areas most affected by COVID-19 [43].

Similarly, the number of trips is also associated with a lower COVID hospital accessibility index and less economic disadvantage, albeit with a population having some type of disability (auditory, visual, or psychomotor). However, as the pandemic grew and reached the distant periphery (with greater marginalization), the demand for medical care increased, which led to the overload of COVID hospitals.

## 4. Discussion

### 4.1. Reflections on the Findings

To confront the pandemic, Mexico employed a COVID hospital strategy that consisted of converting a set of hospitals to provide care exclusively for those infected by COVID-19. The evidence examined in the present study allows us to recognize that the territorial logic on which the COVID hospital strategy was designed has a rational basis. However, the location of the hospitals alone was not enough, since implicitly, increased importance was given to the location, and the importance of coverage and accessibility was dismissed. In other words, for a user, living less than 6 km from a COVID hospital with 40 beds is not the same as living 20 km from one with 450 beds.

In terms of accessibility [44,45], a hospital with 450 beds (compared to a 40 bed hospital) is more accessible for a user, even if it is geographically farther away. In this case, the coverage capacity, which is understood as the number of hospital beds, is decisive in favoring or restricting access to medical care. The crux is the exponential increase in the number of infections that occurred within a short period of time, which caused hospital overload and left large sectors of the population without medical care.

Regarding the general factors that contribute to the spread and multiplication of COVID-19 cases, given that the central city of the MCMA is the area with the highest concentration of jobs and population density, and generates the greatest mobility in the metropolitan area, the results of the present study confirm a number of previous findings on the factors that increase the probability of contagion, such as the size of the city [8] and the density and mobility of the population [9,10,11].

The pre-existing contagion mechanisms, in the case of the MCMA, interacted as follows. As the pandemic worsened and the speed and number of cases increased, the demand for medical care increased exponentially, to the point that a number of medical units became overloaded, which prompted the population to travel greater distances in search of medical care. This was the case in Brazil [46] and in England and Wales [47]. The following question arises from this phenomenon: what general demographic and socioeconomic characteristics distinguish the population that sought medical care for COVID-19 and on which the CMU strategy was based?

In the case of the MCMA, the population sectors with greater urban marginalization distributed in the urban periphery reported the lowest accessibility to COVID-19 hospitals and the need to travel greater distances to obtain medical care. Previous studies have identified the correlation between high vulnerability to COVID-19 and adverse socioeconomic conditions or high marginalization in countries such as Brazil [48] and Mexico [21] and, in specific cases, in places such as Colorado, USA [49], Lima, Peru [50] and the MCMA itself [51].

Similarly, the correlation between low accessibility to COVID hospitals and high marginalization is reflected in higher rates of lethality for this sector of the population [52]. This correlation was also recognized in the case of large cities in developing countries, where the lowest income sectors of the population not only have the lowest accessibility to second- or third-level care hospitals but to outpatient medical care in general [53].

There are multiple lessons that can be derived from the ongoing experience with COVID-19. One is the need to design a territorial strategy for clinics and hospitals at the national level to prepare health services and avoid repeating the mistakes of the past. The magnitude of the damage that this pandemic has caused (and continues to cause) is irrefutable evidence of this need and the importance of addressing it.

The assessment of the location of medical units should begin to occupy a prominent place in the field of public health research [54] and should not focus only on the transfer of patients in emergency situations [55,56,57].

On the other hand, options should be sought to reduce face-to-face presence in hospitals, such as via the creation of virtual medical units, an example being the one that has been launched in CDMX [58]. Another lesson to be learned is the urgent need to strengthen health services by establishing standards of care that are applied in all hospitals, in addition to the generation of interoperability programs not only within a single hospital system but among the various systems of care. The threat is latent, and we must be prepared.

### 4.2. Potential of the Analysis and Future Work

The methodology proposed here to evaluate the CMU strategy in the MCMA is empirically and statistically robust and can be replicated on different scales—city, metropolitan area, state, and region—both in Mexico and in other countries. In addition, the methodology followed sets a precedent for examining the hospital-coverage strategy for other types of potential epidemics, such as dengue, Zika or chikungunya.

Furthermore, this methodology can be used to develop prognostic studies and prepare health services for the impact of potential natural hazards such as earthquakes, hurricanes, floods, volcanic eruptions, or anthropogenic risks, such as explosions and toxic spills at industrial sites, or the collapse of infrastructure or urban equipment, which, when combined with the general demographic and socioeconomic characteristics of the population implies its high level of vulnerability.

## 5. Conclusions

Based on the methodology used in this study to evaluate the CMU strategy in the MCMA, it is concluded that (1) the geographical pattern of COVID-19 contagion moved from the metropolitan center to the urban periphery; (2) as the number of cases increased, overloading medical units, the infected population had to travel greater distances in search of medical care; (3) at the local level, 9 out of 10 inhabitants of the MCMA had a CMU within 7 km; and (4) at the metropolitan level, the population residing in the urban periphery, which has higher levels of marginalization, had the lowest accessibility to CMUs.

The COVID-19 pandemic has been an extreme situation that has revealed unequal accessibility to medical units for different sectors of the population. This points to the need to consider events of this magnitude in the planning of medical services, including the spatial distribution of hospitals, or better transportation infrastructure in the city that may provide equal accessibility to hospitals for the whole population. Future research is needed to address these topics.

## Figures and Tables

**Figure 1 ijerph-19-00665-f001:**
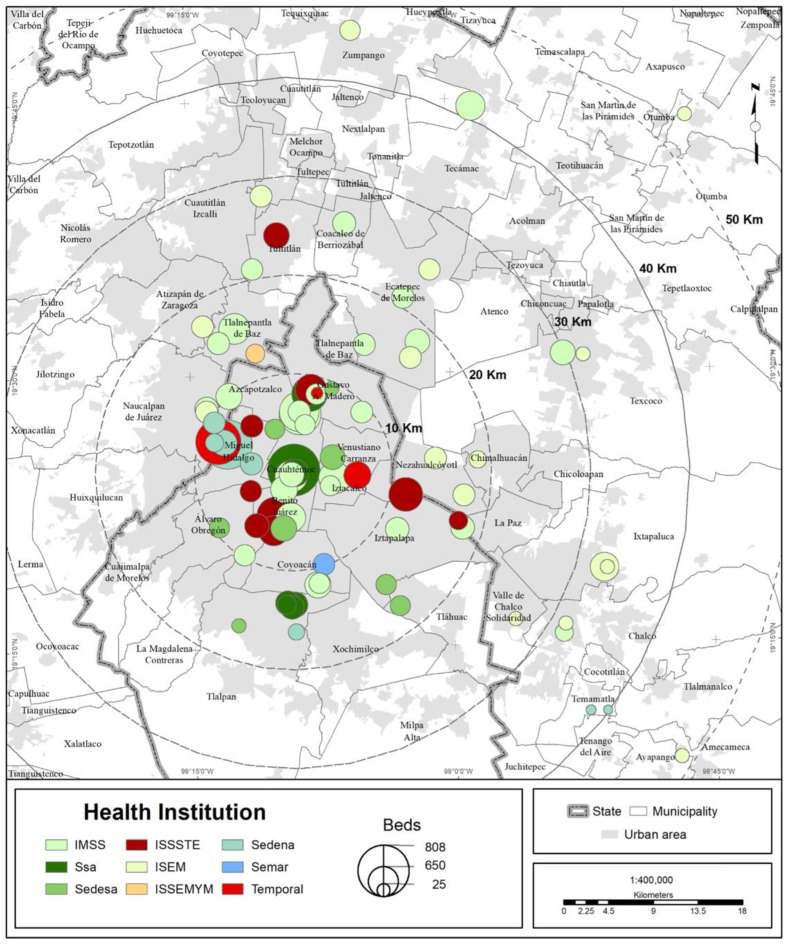
MCMA: distribution of COVID hospitals and total beds.

**Figure 2 ijerph-19-00665-f002:**
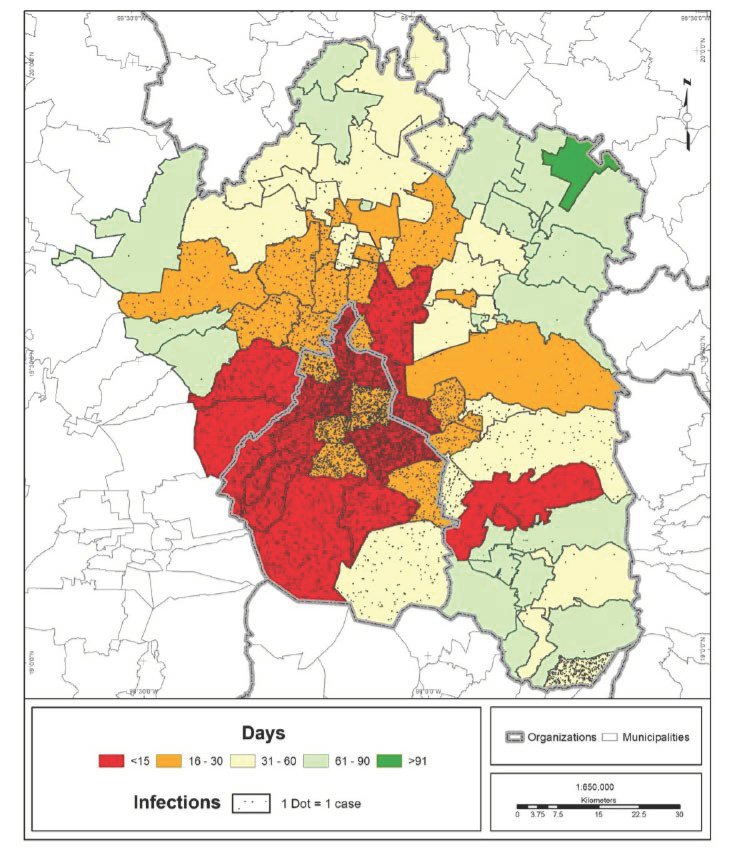
MCMA: days elapsed between the index case in CDMX and the first case in each municipality (30 April 2020). Prepared by the authors with data from [7,28].

**Figure 3 ijerph-19-00665-f003:**
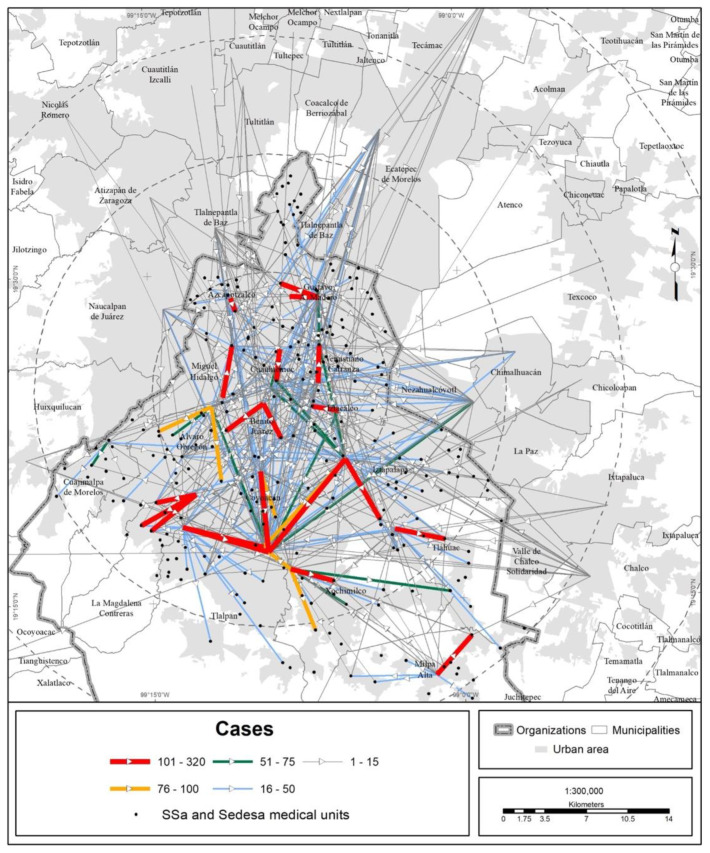
MCMA: trips made to Ssa and Sedesa units due to COVID-19 (data as of 30 April 2020). The origins are the municipal and borough seats; the destinations are medical units. Prepared by the authors with data from [7].

**Figure 4 ijerph-19-00665-f004:**
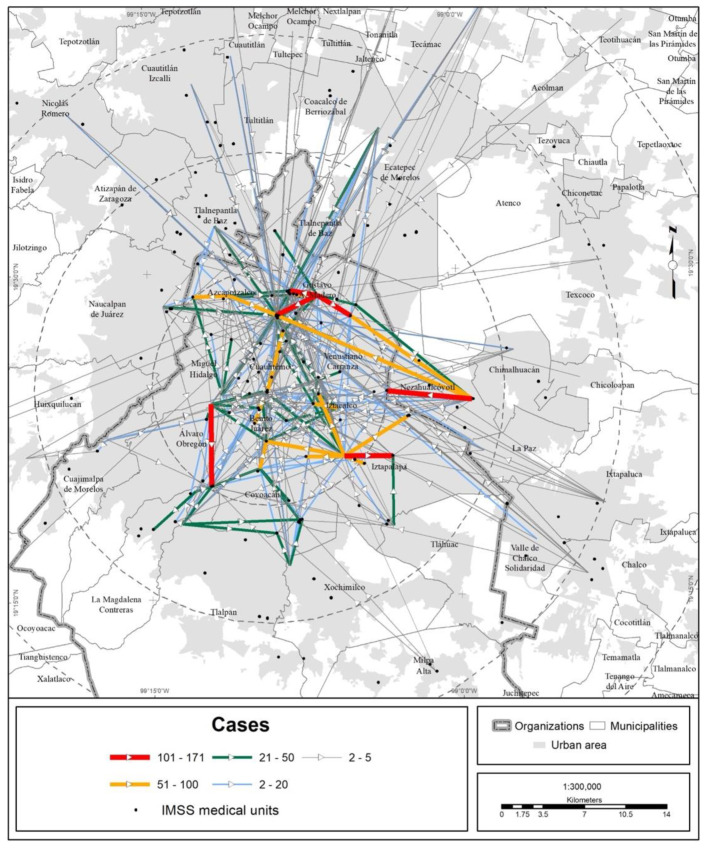
MCMA: trips made to the IMSS unit due to COVID-19 (data as of 30 April 2020). The origins are the municipal and borough seats; the destinations are medical units. Prepared by the authors with data from [7].

**Figure 5 ijerph-19-00665-f005:**
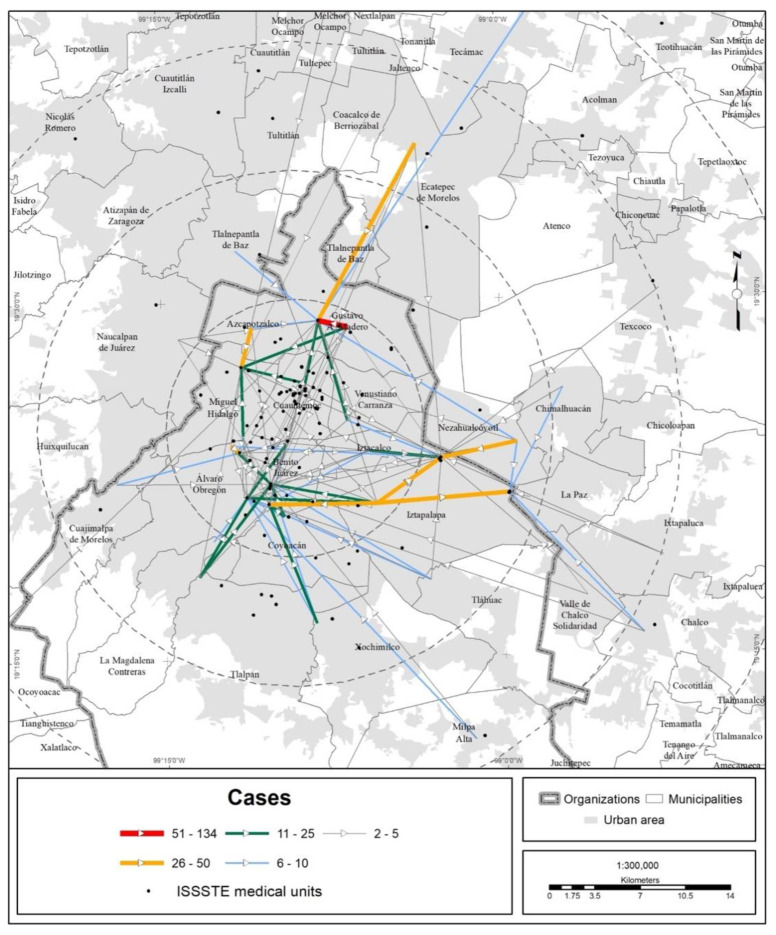
MCMA: trips made to ISSSTE units due to COVID-19 (data as of 30 April 2020). The origins are the municipal and borough seats; the destinations are medical units. Prepared by the authors with data from [7].

**Figure 6 ijerph-19-00665-f006:**
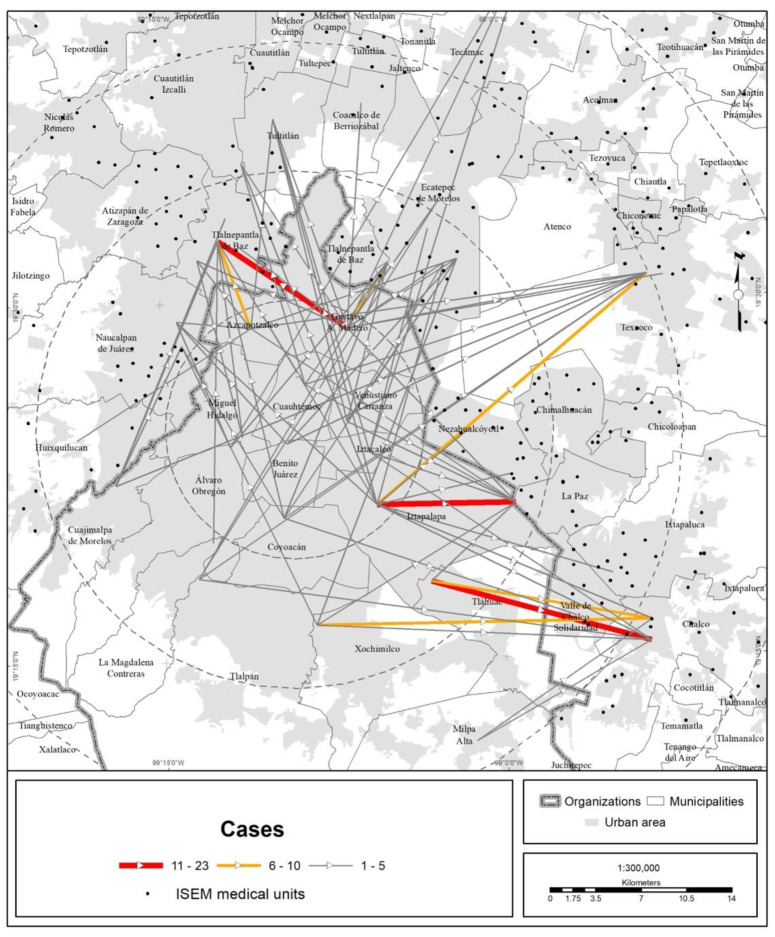
MCMA: trips made to ISEM units due to COVID-19 (data as of 30 April 2020). The origins are the municipal and borough seats; the destinations are medical units. Prepared by the authors with data from [7].

**Figure 7 ijerph-19-00665-f007:**
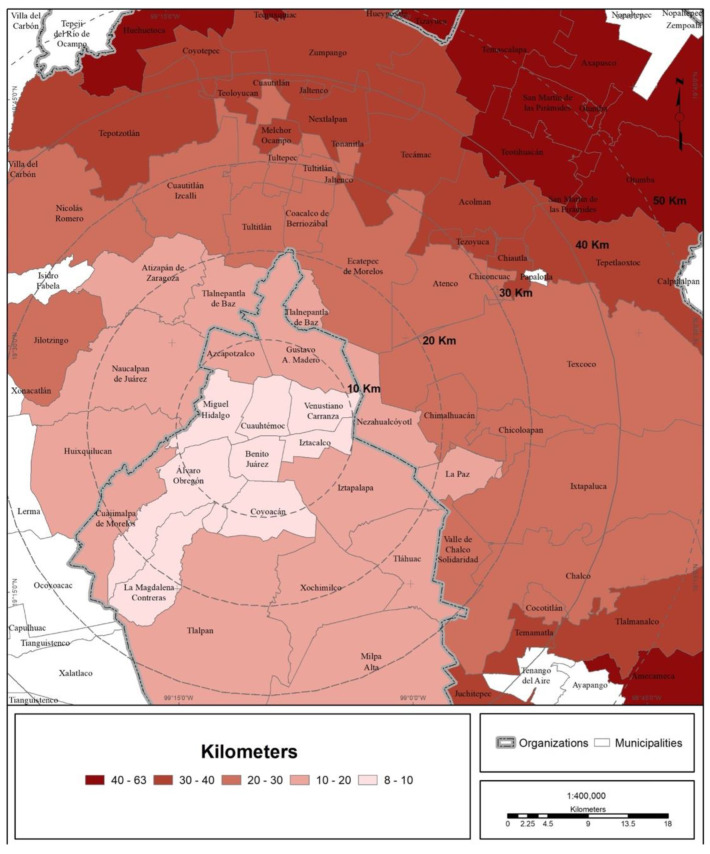
MCMA: average distance to medical units sought because of COVID-19 infection. (Data as of 30 April 2020). Prepared by the authors with data from [7].

**Figure 8 ijerph-19-00665-f008:**
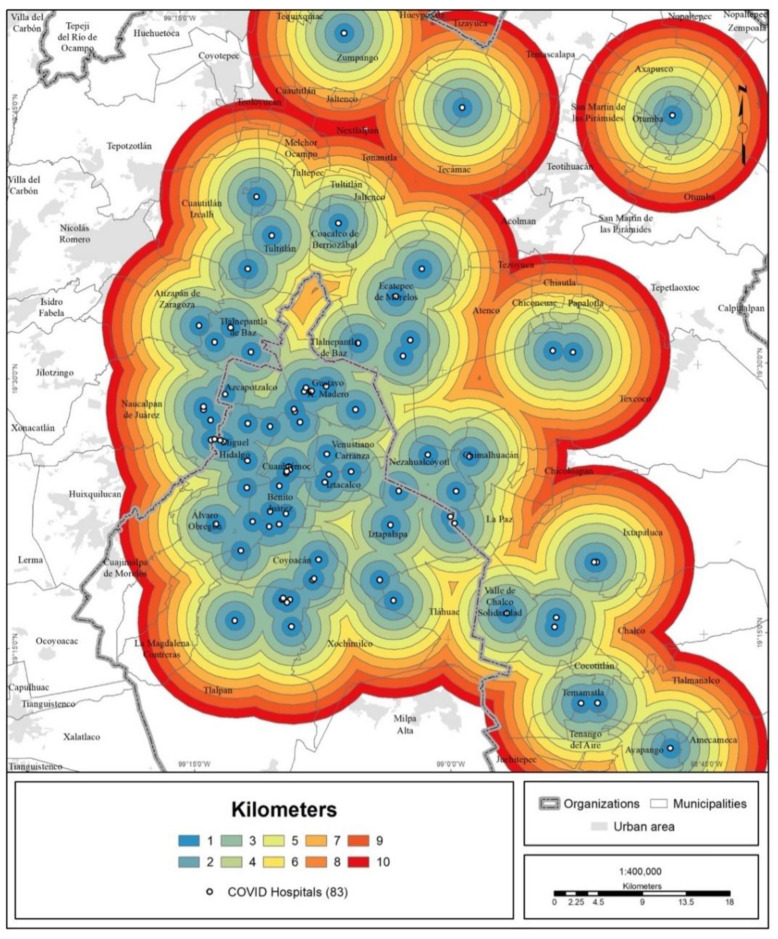
MCMA: coverage areas of COVID hospitals.

**Figure 9 ijerph-19-00665-f009:**
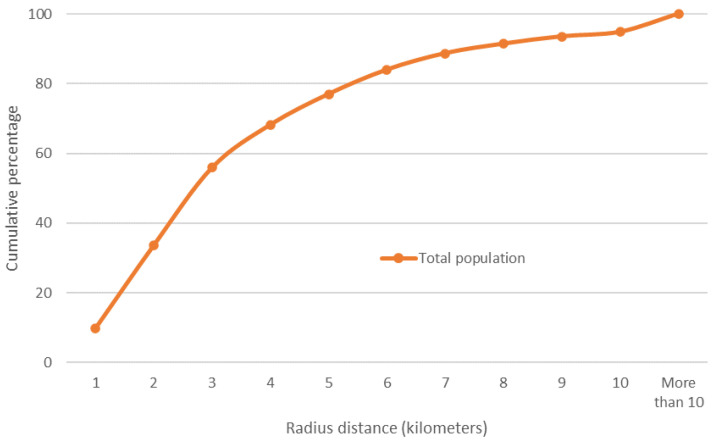
MCMA: cumulative percentage of the total urban population (2020) to distance (radius) from nearest hospital. Prepared by the authors with data from [28].

**Figure 10 ijerph-19-00665-f010:**
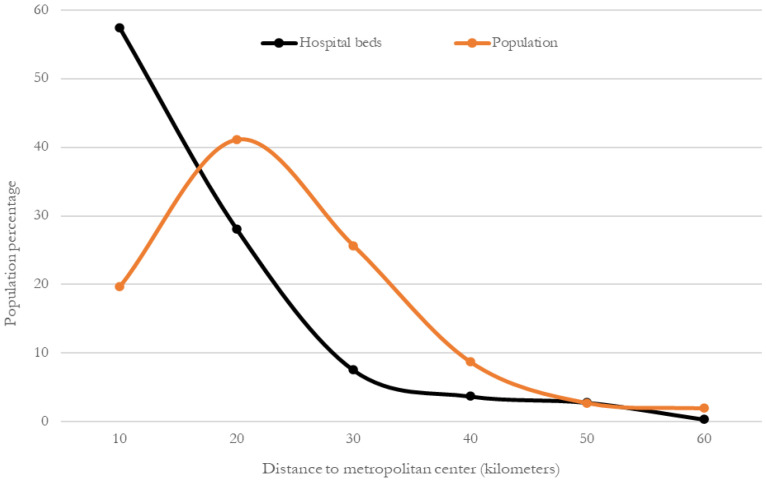
MCMA: percentages of hospital beds and total urban population (2020) to distance from the metropolitan health subcenter. Prepared by the authors with data from [28].

**Figure 11 ijerph-19-00665-f011:**
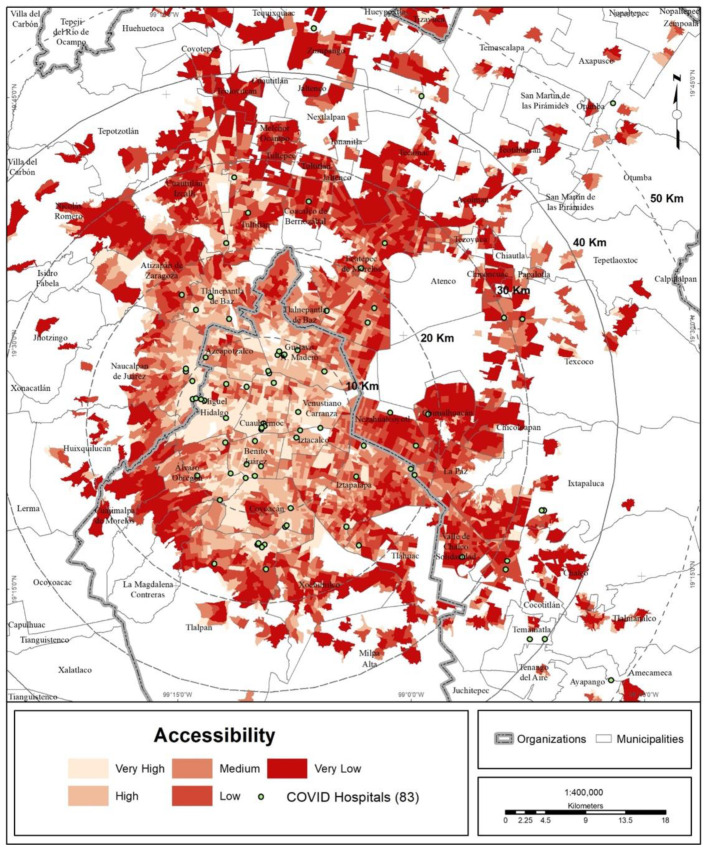
MCMA: accessibility to COVID hospitals.

**Figure 12 ijerph-19-00665-f012:**
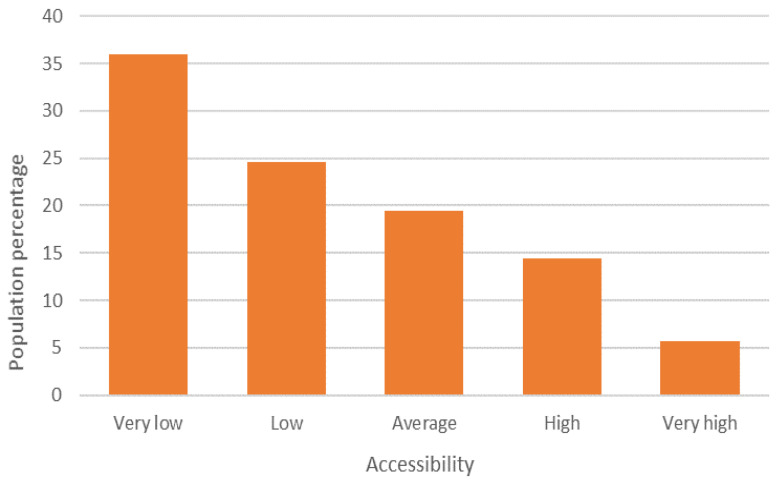
MCMA: concentration of the total urban population (2020) and degree of accessibility to COVID hospitals. Prepared by the authors with data from [28].

**Figure 13 ijerph-19-00665-f013:**
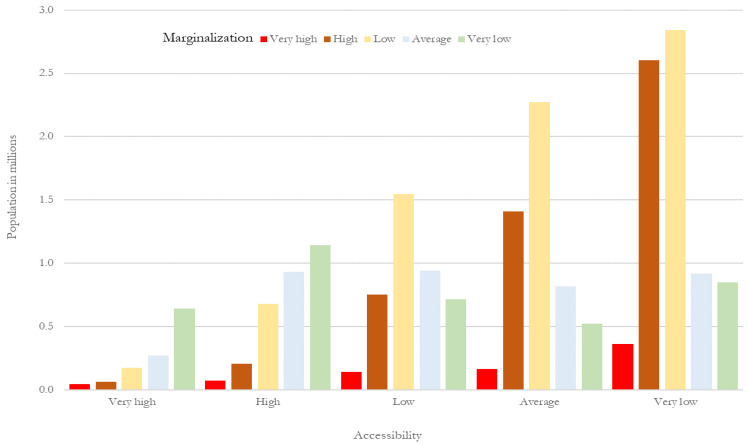
MCMA: percentage distribution of the population according to level of accessibility and level of marginalization. Prepared by the authors with data from [28].

**Table 1 ijerph-19-00665-t001:** MCMA: COVID hospitals and number of hospital beds by institution.

Institution	Hospitals	Beds
IMSS	30	6002
Ssa	7	2050
ISSSTE	9	2044
Sedena	8	1448
Semar	1	140
Sedesa	8	984
ISEM	16	1820
ISSEMyM	1	107
Temporary centers	3	865
Total	83	15,460

**Table 2 ijerph-19-00665-t002:** Linear regression: COVID-positive cases by borough/municipality (as of 30 April 2020).

	Estimate	Std. Error	t Value	Pr (>|t|)
(Intercept)	5.533308	0.841046	6.579	6.54 × 10^−9^ ***
(NL) Days elapsed between the metropolitan index case and the first case in the municipality	−0.643523	0.16859	−3.817	0.000283 ***
Road distance to the metropolitan center (km)	−0.037074	0.011311	−3.278	0.001613 **
Urban population density (population/Ha)	0.016093	0.004587	3.509	0.000781 ***

Signif. Codes: 0 ‘***’, 0.001, ‘**’. Residual standard error: 1.101 on 72 degrees of freedom. Multiple R-squared: 0.7639, Adjusted R-squared: 0.754. F-statistic: 77.65 on 3 and 72 DF, *p*-value: <2.2 × 10^−16^.

**Table 3 ijerph-19-00665-t003:** MCMA: COVID-19-related visits to medical units by health institutions (data as of 30 April 2020).

Institution	Cases	
	Total	%
Ssa and Sedesa	13,878	56.6
IMSS	6878	28
Private institutions	2108	8.6
ISSSTE	1056	4.3
Pemex	237	1
Semar	226	0.9
Sedena	115	0.5
ISSEMyM	16	0.1
Red Cross	5	0.02
IMSS-Opportunities	3	0.01
Other	1	0.004
Total	24,523	100

Prepared by the authors with data from [7].

**Table 4 ijerph-19-00665-t004:** MCMA: average distance traveled to reach a medical unit by health institution (data as of 30 April 2020).

Institution	Average Distance Traveled (Linear Kilometers)
Ssa and Sedesa	16.7
IMSS	14.7
ISSSTE	14.2
ISEM	19.7

**Table 5 ijerph-19-00665-t005:** MCMA: distribution of COVID hospitals and total hospital beds by coverage radius.

Radius (Kilometers)	Total Hospitals	Hospital Beds		2020 Population	
		Total	%	Total	%
10	36	8869	57.4	4,100,074	19.7
20	28	4345	28.1	8,545,588	41.1
30	8	1170	7.6	5,339,309	25.7
40	7	578	3.7	1,823,406	8.8
50	3	438	2.8	575,747	2.8
60	1	60	0.4	409,742	2
Total	83	15,460	100	21,088,201	100

Prepared by the authors based on INEGI, 2021.

**Table 6 ijerph-19-00665-t006:** MCMA: percentage distribution of the population according to level of accessibility and level of marginalization.

		Marginalization					
		Very High	High	Average	Low	Very Low	Total
Accessibility (frequency)	Very high	42,557	64,271	174,556	270,705	641,583	1,193,672
	High	71,191	207,516	679,357	931,892	1,144,483	3,034,437
	Average	140,828	752,870	1,549,083	941,651	716,434	4,100,866
	Low	163,767	1,407,775	2,272,995	816,578	521,055	5,182,168
	Very low	360,166	2,605,597	2,842,277	918,298	850,719	7,577,058
	Total	778,508	5,038,028	7,518,267	3,879,123	3,874,273	21,088,201
		Very high	High	Average	Low	Very low	Total
Accessibility (percentage)	Very high	5.5	1.3	2.3	7	16.6	5.7
	High	9.1	4.1	9	24	29.5	14.4
	Average	18.1	14.9	20.6	24.3	18.5	19.4
	Low	21	27.9	30.2	21.1	13.4	24.6
	Very low	46.3	51.7	37.8	23.7	22	35.9
	Total	100	100	100	100	100	100

Prepared by the authors with data from [28].

**Table 7 ijerph-19-00665-t007:** MCMA: average distance to COVID hospitals by degree of accessibility and marginalization.

Range	Average Distance in Kilometers			
	To the Nearest COVID Hospital		To the Urban Health Sub Center	
	Accessibility	Marginalization	Accessibility	Marginalization
Very high	3.7	7.5	18.5	32.6
High	3.1	5.6	16.4	27.2
Average	3.8	3.7	19.7	19.4
Low	4.3	2.9	23	16.3
Very low	5.8	2.3	28.5	13

**Table 8 ijerph-19-00665-t008:** Linear regression: trips by borough/municipality in search of medical care for COVID-19 (as of 30 April 2020).

	Estimate	Std. Error	t Value	Pr (>|t|)
(Intercept)	2.26736	1.71959	1.319	0.191619
Urban marginalization index	−2.27127	0.48127	−4.719	1.18 × 10^−5^ ***
% population with some disability	0.82066	0.2287	3.588	0.000613 ***
% economically dependent population	−0.98423	0.32194	−3.057	0.003162 **
Road distance to the metropolitan center (kms)	−0.07428	0.01187	−6.259	2.72 × 10^−5^ ***
COVID hospital accessibility index	−1.78524	1.06106	−1.683	0.096923.

Signif. Codes: 0 ‘***’ 0.001 ‘**’ 0.01. Residual standard error: 1.177 on 70 degrees of freedom. Multiple R-squared: 0.8165 Adjusted R-squared: 0.8033. F-statistic: 62.28 on 5 and 70 DF *p*-value: < 2.2 × ^−16^

## Data Availability

The data that support the findings of this study are available upon request to the corresponding author.

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
