# Peer review of "Territorial Strategy of Medical Units for Addressing the First Wave of the COVID-19 Pandemic in the Metropolitan Area of Mexico City: Analysis of Mobility, Accessibility and Marginalization"

_ijerph, 2022, doi:10.3390/ijerph19020665_

Round 1

Reviewer 1 Report

In this paper, there are a couple of issues that the authors should address and revise.

  1. One of the biggest issues is a structure. This paper does not follow the general structure of research articles. Introduction, Literature Review, Materials and methods, etc. Specifically, I don’t still understand why you made the second section. Some parts need to be the introduction or the materials and methods. Also, a lack of efforts on the literature reviews may be responsible for the loose connections in regard to the research problems and necessity. So, you may need to re-structure the manuscript and put more efforts on the literature reviews.
  2. In the introduction, the current problems, research objectives and questions need to be clearly explained. In the current form, the research problems are little vague. In addition to the shortage of clinics and hospitals, you may need to put more explanations about why it matters to explore relations of city’s size, population, and mobility. The research objectives and questions are also not that much clear.
  3. Also, introduction should have a brief summary of the all contents that the manuscript has. For example, the COVID-19 infection trend is stated in Section 4, but it was not introduced in the introduction.
  4. I think that section 2, The spread of COVID-19 and hospital strategy needs to be a part of introduction. No matter who argued that virus mutation will threaten to us in the future and the COVID-19 is one of them. The current situation is that we all are under the pandemic. The hospital shortage problems come from the spread of COVID-19. Also, the explanations about the MCMA and hospital strategies of Mexico would be a part of study area and data section.
  5. Why does the COVID-19 infection trend matter? I am not sure the meaning of the days elapsed between the index case in CDMX and number of cases (Figure 2).

Author Response

Thank you for reviewing the paper, and for the comments. We have addressed all the comments, and are sure that the paper is now better structured and more clear.

1)    As suggested by the reviewer we have merged section 2 with the introduction which was also restructured. We specified the objectives of the paper to make them more clear.
2)    We added a paragraph to the introduction that summarizes the content of the paper.
3)     We added new references to the literature review which we kept as part of the introduction.
4)    We added a small paragraph after Figure 2 in order to make clear the importance of the infection trend:
In an early stage of the pandemic cases appeared in central places where large hospital infrastructure is located, which implies a high accessibility to COVID hospitals in that initial stage of the pandemic. Accessibility  will necessarily decrease as the number of positive cases spread to the periphery of the city.
We also changed the name of Figure 2 in order to make clear that it shows days elapsed since the index case of the city and the first case to appear in each municipality.

Reviewer 2 Report

  1. The stated aim of the manuscript is to evaluate the strategy of the MoH (both Federal and local) implemented to provide healthcare for those affected by the COVID-19 pandemic. If that is the case, it is essential to clarify what type of evaluation authors are proposing and how this evaluation will be implemented. After reading the manuscript, it seems to me that authors may want to evaluate the strategy from an equity perspective, but that is not explicitly stated. In order to evaluate the strategy, it seems to me that it could be relevant to identify what could be an alternative strategy or the reference / recommended strategy.
  2. Given that the second wave hit the country (and the city) harder, it is not clear why the analysis only covers until August 2020. It seems to me that an evaluation of the strategy should cover the period when the demand was higher. 
  3. Description of formula in section 3.3.4 includes Pi that is not in the formula. The rationale on the parameter of impedance —and its value— should be explicit. 
  4. It is unclear the meaning of the number of movements to medical units (are visits to medical facilities?) and how that was measured. 
  5. Some of the reported results seem beyond the scope of the paper; for example, those related to the number of cases and spread of the SARS-CoV-2 in the city. 

Author Response

We would like to thank the reviewer for reading the paper and giving us comments to make it better.
We have tried to address all of the comments the best way possible, and provide arguments for the cases in which we believe the paper should be kept in its current state.

1.    In a new restructured  introduction we have stated the objectives in a clearer way in order to address this comment. Also a paragraph was added to the conclusions section with new research that should follow this paper but that at the moment is outfitted of the scope of the paper.
2.    We suggest changing the title of the paper to specify that we only look at the first wave, as the contagion pattern was spreading from the center of the city to the periphery. A look at the second wave data did not seem to add any new information, but adding it to the current paper would imply much more analysis time.  Furthermore the conversion strategy was implemented during the first wave of the pandemic.
3.    We have changed the formula so that instead of the impedance coefficient being  -λ we use -1. This means that as distance increases accessibility will decrease proportionally, all else being equal.
4.    We changed the word “movement” to “visits to medical units” for greater clarity
5.    We report cases and spread since it is actually the number of cases and the spread of SARS CoV-2 that feeds the analyses done in this research.

Reviewer 3 Report

Very timely and relevant report on a strategy to tackle logistic issues in Healthcare due to COVID. For readers outside of Mexico, the many references to the local health system, abbreviations (of the institutions), maps, etc may be hard to grasp though. Some suggestions:

  • Please explain on the three levels of the healthcare system now mentioned in 1st line of page 2 - what are these levels?
  • Also, please explain on the 'institutions' which include multiple hospitals
  • Please provide a definition and further explanation (how to interpret) of the term 'urban marginalization'
  • Consider moving some of the many figures and tables to the appendix, as the report is now rather extensive - for instance, the relevance of having Figure 3-6 in as 4 separate graphs because they are split by institution is not clear to me
  • page 18, section 5.2 - what is meant by 'a scenario of high vulnerability' and how would it be used? You mean you could identify vulnerabilities? That would make more sense but phrased a bit cryptically now 

Author Response

We would like to thank the reviewer for reading the paper and giving us comments to make it better.
We have addressed all comments suggested, or provide an argument where we believe that the paper should be kept as is.

1)    We added a paragraph in page 4 explaining the three levels of the health care system. This also addresses comment 2.
3)    We have added a paragraph in page 4 at the en of the introduction that explains the marginalization index 
4)    We agree that the number of figures is large, however it is impossible to put all the information of figures 3-6 in one sole map because it is too much information. 
5)    In order to make the sentence more clear we rephrased it: “[…] implies high levels of vulnerability”

Round 2

Reviewer 1 Report

Thank you for your revision. I enjoyed reading the revised version of the manuscript. 

Author Response

We would again like to thank the reviewer for the comments on the paper.